# Nutrient Supplements for Young Children and Mothers’ Self Medication with Over-the-Counter Drugs During the COVID-19 Pandemic

**DOI:** 10.3390/nu16234182

**Published:** 2024-12-02

**Authors:** Esin Aydın Aksoy, Bahar Güçiz Doğan, Sıddıka Songül Yalçın

**Affiliations:** 1Department of Social Pediatrics, Institute of Child Health, Hacettepe University, Ankara 06230, Türkiye; 2Department of Pediatrics, Sisli Hamidiye Etfal Training and Research Hospital, Istanbul 34418, Türkiye; 3Public Health Institute, Hacettepe University, Ankara 06230, Türkiye; bdogan@hacettepe.edu.tr; 4Division of Social Pediatrics, Department of Pediatrics, Faculty of Medicine, Hacettepe University, Ankara 06230, Türkiye; 5Department of Vaccine Studies, Institude of Vaccine, Hacettepe University, Ankara 06230, Türkiye

**Keywords:** COVID-19 pandemic, children, nutrient supplement, over-the-counter drug

## Abstract

Background: The restriction of access to health services during the COVID-19 pandemic has led to an increase in self-medication. This study aims to examine mothers’ use of nutrient supplements with over-the-counter (OTC) medications for their children, including instances of self-medication for themselves. The study also explores maternal characteristics associated with this behavior, the specific medications used, and the reasons for use. Method: In this descriptive study, 450 mothers with children aged 2 to 6 years in Türkiye were recruited through social media platforms. Questions focused on whether mothers used supplements for themselves and their children, types of products, frequency, and reasons for use. Multivariable binary logistic regression was conducted to examine the factors associated with OTC medication use for children. Result: Nearly half of the mothers reported administering OTC medications to their children. Factors associated with this practice included the child’s age (specifically 48–72 months), attendance at nursery, perceived underweight status, and regular health visits with a pediatrician. Additionally, mothers who frequently used medications without a doctor’s recommendation were 5.8 times more likely to give OTC drugs to their children. Maternal self-medication was significantly associated with an increased likelihood of OTC medication use for children (OR = 12.1). The most commonly used supplements included vitamin D, fish oil, multivitamins, vitamin C, immune boosters, zinc, probiotics, herbal teas, oral/nasal sprays, throat lozenges, and aspirin, with the primary purposes being prevention and treatment. Conclusions: The administration of OTC medications in young children, who rely heavily on maternal care should be more closely monitored to ensure their safety and well-being, especially during epidemics.

## 1. Introduction

During the COVID-19 pandemic, the imposed lockdown and increasing numbers of COVID-19-related deaths created panic among the general public. Many individuals turned to electronic media and social networking to obtain the latest information about the disease and to seek any positive news regarding treatment and prevention efforts [1,2,3]. However, the uncontrolled spread of misinformation through electronic and social media led many people to self-medicate [4]. Additionally, restrictions on access to medical care during the pandemic further fueled this trend. Many individuals perceived doctors and nurses as potential carriers of the virus, making them hesitant to visit hospitals. Consequently, the inclination to self-medicate increased significantly [5,6]. It is believed that the notion of repurposing long-established medications as potential cures or preventive treatments for COVID-19 is largely driven by media narratives [4,5,7].

According to the World Health Organization, self-medication is a significant aspect of self-care and is defined as the use of medications to address self-diagnosed conditions or the self-administration of medications prescribed by a doctor for chronic or recurring diseases and symptoms [8]. Self-medication is also defined as taking medication on one’s own initiative or on the initiative of someone who is not medically qualified [9]. Responsible drug use refers to the use of over-the-counter drugs (OTC) according to the package instructions for use or the doctor’s prescription. Appropriate self-medication may have benefits, such as greater access to medication and allowing patients to take an active role in their own healthcare [10]. However, self-medication is far from a completely safe practice. The potential risks of self-medication include incorrect self-diagnosis, delay in seeking medical attention when needed, infrequently detected but severe adverse reactions, dangerous drug interactions, incorrect routes of administration, incorrect dosages, incorrect choices of therapy, masking of a serious illness, and addiction and abuse risk [10,11]. Inappropriate self-medication also includes taking prescription medicine without a prescription, using old medicines prescribed for other illnesses, sharing drugs with friends/family, and using expired drugs [6,12]. Even though prescription and non-prescription medicine use is being well researched in developed countries, there are only a few studies focused on developing countries. In Türkiye, compared to 2008, non-prescription medicine use decreased until 2012; however, an increasing trend appeared after 2012 [13]. In Türkiye, the Ministry of Health is the authority that decides whether pharmaceuticals should be sold with or without a prescription and their prices, but the reimbursement list for pharmaceuticals is determined by the Social Security Institution. There is no legal regulation in terms of non-prescribed medicine use in Türkiye [13].

Ensuring good nutritional status, including a healthy body weight, is a reasonable approach in the prevention of the viral infection SARS-CoV-2 or in alleviating its course [14]. The body’s immune response is formed by the interaction of vitamins D, C, and A; zinc; selenium; and polyunsaturated n-3 fatty acids and probiotics. During the pandemic period, they played an important role in maintaining the integrity and functions of the immune system, including the activation, differentiation, and proliferation of immune cells and the preservation of the stability of cell membranes [14,15].

There are few studies investigating how the COVID-19 pandemic has influenced the intake of supplemental food and OTC medications [4,5,7,16]. In this study, we aimed to examine mothers’ use of OTC for their children, including self-medication for themselves. We hypothesize that maternal OTC medication usage will increase in response to factors such as health concerns related to the pandemic, the reduced accessibility of healthcare services, and specific socio-demographic characteristics.

“The Theory of Planned Behavior” offers a valuable theoretical framework for understanding self-medication [17]. This theory posits that an individual’s intention to engage in a behavior is influenced by three key factors: attitudes toward the behavior, subjective norms, and perceived behavioral control [17]. In the context of self-medication, mothers’ attitudes toward using OTC medications for themselves and their children—shaped by their beliefs about efficacy and safety—play a critical role in their decision-making process [17]. Additionally, the influence of social norms, such as advice from family and friends or societal perceptions of self-medication, can significantly impact mothers’ behaviors [17]. Lastly, perceived behavioral control refers to mothers’ beliefs about their ability to obtain and administer medications safely and effectively [18]. By applying this framework, one can gain insights into how these factors contribute to self-medication practices among mothers during health crises like the COVID-19 pandemic.

Upon the completion of this study, we expected to gain valuable insights into the factors associated with OTC medication use in children, as well as broader patterns of self-medication among mothers. These findings could inform public health strategists aimed at promoting safe OTC usage, particularly during health crises like the COVID-19 pandemic.

## 2. Materials and Methods

### 2.1. Study Design and Population

In this descriptive study, mothers with children aged 2 to 6 years in Türkiye were recruited through social media platforms such as Instagram, Facebook, WhatsApp, Telegram, and email by sharing a web-based questionnaire conducted from April 2021 to July 2021. The survey link was shared via WhatsApp groups where mothers were members and on social media platforms like Instagram using the snowball sampling method. The first page of the Google survey contained an information note and maternal consent. Participants gave their consent by ticking the option stating “I agree to participate in the study”. Mothers who gave consent proceeded to the second page to answer the questions. The study protocol was approved by the Hacettepe University Scientific Research Board.

The inclusion criteria for the mother–child pairs were as follows: the child had to be between 2 and 6 years old, and the mother had to be the primary caregiver, able to use social media, and willing to provide informed consent.

### 2.2. Sample Size

Using the OpenEpi program [19], it was determined that 384 participants would be needed to calculate a condition with a 50% prevalence at a 5% confidence level, assuming a design effect of 1. Anticipating that 25% of the surveys might be filled out incorrectly, the plan was to collect 480 participants.

### 2.3. Survey Forms

The survey form consisted of 37 questions. The first 22 questions focused on socio-demographic characteristics, including family details (such as the age, education level, and occupation of parents; household income; family structure; maternal social media use; and type of residence) and child-specific characteristics (age, sex, birth order, duration of breastfeeding, the presence of any self-reported illness, history of COVID-19 infection, routine child health checkups, and nursery school attendance during the pandemic). Additionally, mothers were asked about their perceptions of their children’s weights and heights.

The survey also included 15 questions focused on supplement use for themselves and their children, the types of products used, frequency of use, and the reasons for use. If they participated in supplement use, they could select all appropriate options from the following list: vitamin D, vitamin C, multivitamin formulations, fish oil, propolis, probiotics, zinc, aspirin, immune boosters, herbal tea, mouth and nose sprays, throat lozenges, and others. Finally, the survey explored the mothers’ knowledge of OTC medication side effects.

Maternal and paternal ages were classified into three groups: under 30, in the range of 30–35, and over 35 years. Parental education was grouped into two groups: less than 12 years and 12 years or more. While grouping the occupations, since teachers and healthcare workers were relatively overrepresented in the sample, these categories were evaluated as separate categories, forming four groups: teachers, healthcare workers, others, and unemployed parents. The children’s ages were classified into two groups: 24–47 months and 48–72 months. Family size was grouped into three categories: nuclear family, extended family, and single mother. Breastfeeding duration was examined in five groups, each representing a 6-month interval (mothers who breastfed beyond the age of 2 were categorized as >24 months).

### 2.4. Statistical Analyses

The data were uploaded from web file to the program “IBM Corp. Released 2015. IBM SPSS Statistics for Windows, Version 23.0. Armonk, NY, USA: IBM Corp”. The Chi-square test or Fisher’s exact test was applied to evaluate differences observed in the distribution of the defined variable categories where appropriate. When the Chi-square analysis for the 3 × 2 and 4 × 2 groups denoted significant differences, adjusted residuals and Bonferroni correction were carried out to show comparisons in two groups. Univariate binary logistic regression was used to evaluate the association between OTC medication use for a child aged 2–6 during the pandemic and family–child characteristics.

Variables with a *p*-value < 0.2 in the univariate analysis were included in further analysis. Due to the correlation between maternal and paternal education levels, only maternal education was included in the model. Similarly, since there was a relationship between the mother’s perception of their child’s weight and height. When the Chi-square analysis for the 3 × 2 and 4 × 2 groups denoted significant differences, adjusted residuals and Bonferroni correction were carried out to show comparisons in two groups. Univariate binary logistic regression was used to evaluate the association between OTC medication use for a child aged 2–6 during the pandemic and family–child characteristics., only weight perception was incorporated into the model. A multivariable binary logistic regression [METHOD = ENTER] was conducted to examine the factors associated with OTC medication use for children aged 2–6 during the pandemic (dependent variable). The analysis revealed significant associations with various factors, including residence (city vs. county), maternal age (in years), maternal education (≥12 vs. <12 years), family type (extended vs. nuclear family; single mother vs. nuclear family), parental occupation (at least one parent being a healthcare professional vs. not), perceived family income (high vs. middle/low), child’s age (48–72 months vs. 24–47 months), birth order (firstborn vs. second or later), type of birth (C-section vs. vaginal delivery), attendance at nursery during the pandemic (yes vs. no), history of COVID-19 infection (yes vs. no), the mother’s perception of the child’s nutrition (underweight vs. normal), the presence of routine child health checkups (family health unit vs. none; pediatrician vs. none), maternal use of medications without a physician’s recommendation before the pandemic (sometimes vs. never; often vs. never), and maternal self-medication during the pandemic (yes vs. no). A *p*-value < 0.05 was considered statistically significant.

## 3. Results

By the end of the study period, 492 mothers were recruited. The questionnaires with more than 10% of the questions left partially unanswered (n = 6) and those from 36 children who were outside the defined age range were excluded from the study. As a result, data from 450 mother-child pairs were included in the analysis.

### 3.1. Family–Child Characteristics

The mean age of the mothers was 33.6 years with a standard deviation of 4.7 (range: 22–49 years). One-fifth of the mothers in this study were under the age of 30, while 34.2% were over the age of 35. Two-thirds of the mothers had more than 12 years of education, and almost three out of five worked as healthcare professionals, teachers, or in other occupations. More than half of the fathers were over 35 years old, and nearly three-fourths had more than 12 years of education. Overall, 82.7% of the families were nuclear; 37.5% of the children were the first-born in their family. One in four children had attended nursery school during the pandemic (Table 1).

### 3.2. The Relationship Between OTC Medication Use and Mother–Child Factors

Nearly half of the mothers had given OTC medications to their children. We found in our study that higher parental education levels were associated with more frequent OTC medication use in children (*p* < 0.001). Additionally, when the mother or father was a healthcare professional, the frequency of OTC medication use was higher compared to other groups (*p* < 0.001). Families with higher incomes also reported more frequent OTC medication use compared to middle- and low-income families (*p* < 0.001) (Table 1).

The proportion of OTC medication use was higher in the first-born child than in younger children in families (*p* = 0.003). The percentage of OTC medication use was significantly higher among children born via C-section (*p* = 0.008). There was no statistically significant association between the child’s sex, gestational age, or breastfeeding duration and OTC medication use. However, children aged 48–72 months were given supplementary food more than those aged 24–47 months (*p* = 0.003). OTC drug use was higher among those who attended nursery compared to those who did not, and the difference was statistically significant (*p* < 0.001). The presence of any chronic illness was not associated with supplement use. A higher proportion of children with a family history of COVID-19 infection were administered OTC medications compared to those without such a history (*p* = 0.036). OTC drug use was the highest among children whose routine health supervision was provided by a pediatrician (*p* < 0.001) (Table 1).

The proportion of OTC medication use was also higher in mothers who perceived their children as underweight (*p* = 0.022). In this study, 10.6% of mothers reported using medications for their children without a physician’s recommendation, and 75% of these mothers had given OTC drugs to their children (*p* < 0.001). Additionally, maternal self-medication during the pandemic increased the use of supplements for their children (*p* < 0.001) (Table 1).

Finally, it was found that the mothers’ consideration of OTC drug use for their children after the pandemic was closely related to the proportion of OTC drug use during the pandemic (Table 1).

### 3.3. Types of Supplements

In this study, nearly half of the mothers (45.3%) were using at least one type of supplement for their child. The most commonly used supplement was vitamin D, with half of the mothers who gave supplements choosing vitamin D, representing one-fifth of all the mothers in the study. The second most common supplement was fish oil, used by 20% of the mothers. Multivitamin preparations and propolis were used with similar proportions. Other supplements included vitamin C, immune boosters, zinc, probiotics, herbal tea, mouth and nasal sprays, throat lozenges, aspirin, and various other supplements (Table 2).

When asked about the purpose of using supplements, the majority of mothers responded that it was to prevent illness in their child. Other reasons included treating the child, improving the child’s intelligence, increasing the child’s appetite, and promoting growth. Half of the mothers reported using supplements daily. Notably, 76.5% of mothers had been giving supplements to their children even before the pandemic. Almost all mothers believed that the supplements were beneficial for their child, and half of them had no concerns about potential side effects. One-third of mothers who used supplements had received advice from relatives, neighbors, teachers, or social media (Table 3).

### 3.4. Multiple Logistic Regression Analysis of OTC Medication Use

Variables with a *p*-value < 0.2 in the univariate analysis were used for further analysis (Table 4). In the multiple logistic regression analysis, OTC drug use was found to be 2.04 times higher in first-born children compared to children who were born second or later. Attendance at nursery school during the pandemic increased the likelihood of OTC drug use by 2.29 times. As the mother’s age increased, the use of OTC medications also increased, with the odds ratio increasing by 0.08 for each additional year of age. Additionally, OTC drug use was 2.10 times higher in mothers who perceived their child as underweight compared to those who thought their child’s weight was normal. Compared to mothers who did not use medications without a doctor’s recommendation, OTC medication use was 1.8 times higher in mothers who sometimes used medication without a doctor’s recommendation and 5.76 times higher in mothers who frequently did so. Mothers who took their child to a pediatrician for routine health supervision without any complaints were 2.88 times more likely to give OTC drugs to their child than those who did not. Furthermore, among the maternal self-medication group, OTC drug use was 12.05 times higher than that of the non-medication group (Table 4).

## 4. Discussion

In this study, the proportion of using OTC drugs for children by mothers during the pandemic was 45.3%. Most studies in the literature about using supplements have been conducted on adults during the pandemic [4,5,7,20]. A review study reported the prevalence of self-medication as 44.7% [21]. The prevalence of using OTC drugs as a self-medication technique or using herbal medicines during the COVID-19 pandemic was reported to be 34.3% in Pakistan, 39.3% in Türkiye, 84% in Iran, 99% in Ethiopia, 80% in Indonesia, and 45.6% in Poland [5,22,23,24,25,26].

It was observed that OTC drug use was more common in mothers over 35 years of age in the current study. Similarly, in an another study conducted with 100 mothers during the pandemic period, it was shown that using complementary and alternative medicine (CAM) for their children was more common among older mothers, which might indicate that as the mother’s age increases, more experience with serious illnesses in their immediate family could be gained [27].

In this study, another observation was that the proportion of OTC drug use for the child increased with the increase in parental education. The results in the literature are controversial [21,28,29,30,31,32,33,34,35,36,37,38,39,40]. At higher education levels, people may feel more capable of diagnosing their own illnesses and deciding on treatments for their illnesses. Since the time allocated to the patient in the doctor’s examinations was short, there may be distrust of the doctor, and highly educated people can feel more confident in themselves by reading and understanding the product’s usage instructions [41]. However, there are fewer studies showing that the frequency of OTC drug use increases as the level of education decreases. A study about mother–children couples was conducted in Iran [42], and the participants of two other studies were adults, and the topic of these studies was self-medication with Non-Steroidal Anti-Inflammatory Drugs, antihistamines, and antibiotics [43,44]. Studies have shown that self-medication is more common among individuals with lower education levels, while those with higher education levels tend to be more skeptical, critical, and cautious about self-medication [5].

When comparing our study findings with the literature, we observed that the proportion of OTC drug use was higher among employed mothers [29,32]. This situation could be attributed to the employed mothers’ time constraints and their desire to return to their daily routines as soon as possible [29]. When maternal occupations were grouped as healthcare workers, teachers, and other, healthcare worker mothers were more likely to give supplements to their children. Similarly, a study in 2020 conducted before the pandemic showed that mothers with healthcare professions had self-medicated their children more than the others [40]. All fathers in the families participating in our study were working. The proportion of fathers working in the healthcare field was 6.7%, and most of the children whose fathers were healthcare workers received supplements. A study conducted in 2020 in Türkiye researched the use of dietary supplements and related factors in healthcare workers, and the authors stated that the use of supplement nutrients is more common among healthcare workers than among the public in general [45]. Studies in the literature were generally conducted with mothers. However, in a study conducted in 2018 in India, it was shown that employed fathers had increased use of complementary and alternative medicines for their children [46]. This situation can be attributed to healthcare professionals’ awareness and knowledge of medical issues.

In our study, we found that families with higher incomes had a higher frequency of supplement use for their children during the pandemic. Similar research conducted during the pandemic has also indicated that higher income levels are associated with increased supplement usage [25,26]. Additionally, studies from before the COVID-19 pandemic similarly demonstrated that families with higher incomes were more likely to use supplements [28,29,31,33,34,37,47,48,49,50]. We also found that cesarean deliveries were associated with a higher likelihood of OTC drug use. This finding aligns with a 2015 study on pediatric drug use, which similarly observed that cesarean delivery was linked to an increased likelihood of medication use in children [51].

There are limited studies investigating drug use in children during the pandemic. One study from the USA examined trends in pediatric non-prescription analgesic and antipyretic exposure during the COVID-19 pandemic and found that the use of these medications among children under 6 years old decreased during this period. This decline was attributed to a sharp reduction in influenza, other viral infections, and injuries [52]. In our study, we found that the likelihood of giving OTC medications was higher for children aged 48–72 months compared to those aged 24–47 months. A review of 58 studies involving children aged 0–21 years published in 2014 revealed that in 12 of these studies, the frequency of OTC drug use was significantly higher in older age groups, while only one study reported a higher frequency in younger children [34]. Additionally, a 2014 study of German children found that the use of herbal medicines was more common in those under 6 years of age [47]. Similarly, the 2015 Canadian Community Health Survey indicated that the prevalence of vitamin and mineral supplement use was the highest among children aged 2–5 years and the lowest in children aged 14–18 years [48]. Furthermore, a study examining dietary supplement usage among children in the USA found that the prevalence was the highest in children aged 2–5 years [49].

We evaluated the relationship between maternal and child characteristics and OTC drug use during the pandemic using multiple logistic regression. It was found that mothers were 2.04 times more likely to use supplements for their first-born child than for children born later. This contrasts with a 2014 study from Iran, which showed that having two or more children increased the use of CAM for children [42]. Similarly, a 2022 study from Colombia on COVID-19 found that self-medication with antibiotics increased with the number of children in the family, likely due to the mothers’ prior positive experiences in treating older children with self-medication [53].

In our study, mothers who perceived their child as underweight were 2.1 times more likely to use OTC drugs for their child than those who perceived their child as having a normal weight. Additionally, we observed that the odds for OTC drug use in mothers with children attending a nursery was 2.29 times higher than the mothers of children not in a nursery. No studies have examined the relationship between nursery attendance and OTC drug use during the pandemic; however, pandemic studies focused on the link between school attendance and disease transmission have been conducted [54,55]. Working mothers who sent their children to a nursery during the pandemic may have used OTC drugs to protect their children from COVID-19.

Our findings also show that those who regularly used drugs without a doctor’s recommendation before the pandemic were 5.7 times more likely to use OTC drugs for their children during the pandemic than those who rarely or never self-medicated before. A study with 782 adult participants from Iran found that over half consulted a physician before using dietary supplements like vitamin D, vitamin C, multivitamins, iron, zinc, calcium, and omega-3; however, only 23.2% consulted a physician before using medicinal herbs such as chamomile, ginger, thyme, mint, and cinnamon [22]. A Turkish study during the pandemic found that one-third of participants did not inform their physicians about using traditional and complementary medicine [24]. Similarly, in a study, two-thirds of mothers with school-age children used anti-COVID-19 medications and supplements for themselves without consulting a physician [26].

In a study, 16.9% of adults reported taking prescription drugs during the pandemic without consulting a doctor, and 10.2% had not self-medicated before the pandemic [5]. Conversely, a study involving 18 mothers on their beliefs toward self-treatment showed that self-medication for children did not increase during the pandemic, as these mothers maintained strong relationships with family doctors [56]. Similarly, a study of adults during the COVID-19 pandemic found that self-medication was often practiced due to difficulty accessing healthcare professionals [23]. Although many studies reported an increase in self-medication during the pandemic, most participants were adults, while children—being a vulnerable group—should be carefully monitored for OTC drug use. Our study also found that mothers of children who had routine health check-ups with a pediatrician were 2.88 times more likely to use OTC drugs for their child than mothers of children who did not have routine pediatrician visits. Regular pediatric visits may increase mothers’ familiarity with and access to information about various medications [57].

A 1997 study from Norway involving 5454 women and 5122 men evaluated medication use patterns across a general population aged 0–80 years. The results indicate that women used medication more frequently than men, which was attributed to a higher rate of diagnosed conditions such as diseases, illnesses, and injuries among women [58]. The high rate of maternal self-medication in our study may be influenced by the focus on mothers as participants. If fathers had been included in the study, the self-medication rate might have been lower.

Our study found that the top three supplements used by participating mothers for their children were vitamin D, fish oil, and propolis. Other supplements included multivitamins, vitamin C, immune boosters, zinc, probiotics, and herbal teas. Vitamin D was used by 22.7% of mothers, while fish oil was used by 20%. A study investigating anti-COVID-19 medications and supplements among women reported vitamin C, ginger, and honey as the most frequently used [26]. Another study with adults identified CAMs such as vitamin C, relaxation techniques, prayer, psychotherapy, ginger, and omega 3,6,9 as being commonly used [59]. In Iran, a study of the general population revealed that 61.3% used dietary supplements, 57.9% engaged in prayer, and 48.8% relied on herbal medicines [22]. A study of Chinese adults reported that vitamin C, probiotics, alcohol, and vinegar were commonly included in their dietary practices during the COVID-19 outbreak [20]. In Ethiopia, adults used traditional remedies such as eucalyptus, damakese, ginger, rue, fetto, zingibil, garlic, and lemon tea to manage COVID-19 [25]. Data from Google Trends in Turkiye how that vitamins D and C were the most searched supplements in Turkiye and globally during the pandemic [60]. To our knowledge, this is the only study in the literature specifically assessing supplement use among children.

In our study, four out of five mothers reported that they gave supplements to their children primarily to prevent illness. About one-third of mothers used supplements to treat their child when already sick. Other reasons included enhancing the child’s appetite, promoting growth, and boosting cognitive development. Consistent with these findings, 87.3% of mothers who used OTC supplements in our study did so as a preventive measure before illness onset. Similarly, a study from Indonesia found that 88.3% of women used anti-COVID-19 medications to boost immunity, while other reasons included preventing fatigue, reducing symptoms, supporting daily activities, and combating SARS-CoV-2 [26]. A study from Turkiye analyzed Google Trends data during the COVID-19 pandemic and observed a rise in searches related to “immunity”, with popular phrases being “Strengthening the immune system in children”, “Drugs that disrupt the immune system”, “What to eat to strengthen the immune system”, and “The best drug to strengthen immunity” [60]. Supporting this trend, research in Lithuania indicates that while general supplement use decreased post-pandemic, those who took supplements to strengthen health continued their use [61]. Furthermore, studies have shown that mothers, who often play the primary caregiving role, are the key decision makers in administering medications to children within the family [40]. This highlights the importance of ensuring that mothers have access to reliable, evidence-based information to make informed health-related decisions for their children. Another study on complementary and alternative medicine use during the COVID-19 pandemic found that infection prevention and anxiety reduction were among the main motivations for using CAM [22].

Half of the mothers using OTC medications in this study administered these drugs to their children daily, and two-thirds had been giving supplements even before the pandemic. This routine use might explain non-infectious purposes for supplementation, such as supporting growth, boosting appetite, and enhancing cognitive abilities. Most mothers in this study believed that these supplements were beneficial for their children. Notably, four out of five mothers reported using OTC drugs for their children based on a doctor’s recommendation. In contrast, a study from Poland found that only 8.3% of adults consulted a doctor before self-medicating [5]. Similarly, two-thirds of Indonesian mothers reported using COVID-19-related medications without medical advice [26], and research from Romania indicates that the pandemic did not significantly change the rate of self-medication without a physician’s consultation among mothers [56]. A Turkish study showed that one-third of participants did not disclose their COVID-19-related medication use to their doctors [24], while in an Iranian study, 55% of participants consulted a doctor before using dietary supplements [22]. While there are limited data on children, these findings suggest that parents are generally more cautious and likely to consult a physician regarding their children’s medication use compared to their own.

In our study, half of the mothers who used OTC drugs for their children reported no concern about potential side effects. A study conducted during the COVID-19 pandemic showed that individuals informed about side effects by their physicians used self-medication twice as frequently as those without this information [23]. In another study, over half of the participants believed complementary medicines to have fewer side effects than conventional medicines [24]. While adverse effects can occur with any drug, parents should remain vigilant regarding OTC drug safety, as supplements and “natural” products are not inherently risk-free.

### Strengths and Limitations

This study has several strengths, including being one of the few to examine the association between the pandemic and OTC drug and supplement use in children. It provides a detailed analysis of the relationship between maternal and child characteristics and OTC medication use, which is a valuable contribution to the literature.

Breastfeeding is recommended for up to two years and beyond, but during pandemic periods and when children are sick, continuing breastfeeding becomes even more crucial for supporting child health [6,62,63]. As a limitation, since our study focused on children aged 2–6 years, we did not inquire about their breastfeeding status. In our study, 22.4% of the children were breastfed for 24 months or longer, and 36 (28.8%) of the 126 children aged 24–35 months continued breastfeeding after the age of 2. However, the use of OTC medications was 32.0% among breastfed children compared to 26.7% among those who were not breastfed (*p* = 0.519). This suggests that during the pandemic, the uncertainty surrounding COVID-19 led families to turn to supplementary foods.

Additional limitations include the reliance on self-reported data collected through surveys, which may affect the accuracy of the responses. Another limitation is that a significant portion of participants in our study were healthcare workers and teachers. Additionally, this study is limited to mothers, excluding insights from fathers or other caregivers. Finally, the changing healthcare access conditions during the pandemic may have influenced medication use, making it challenging to fully control for this variable.

## 5. Conclusions

This study highlights the significant factors associated with the use of OTC medications and nutrient supplements among children, a vulnerable group, during the COVID-19 pandemic, with a particular focus on maternal behaviors. Our findings indicate that higher-income families tended to use supplements more frequently for their children and that mothers were more likely to resort to OTC medications, particularly for their first-born child. Notably, mothers who perceived their children as underweight or those whose children attended a nursery were more inclined to use OTC drugs. The results suggest that a majority of mothers utilized supplements primarily for preventive purposes rather than for treatment, reflecting a strong inclination towards safeguarding their children’s health. While many mothers sought medical advice before administering OTC medications, a concerning proportion also engaged in self-medication without professional guidance. These insights underscore the importance of ensuring that parents have access to reliable health information and guidance on the safe use of OTC medications, particularly during health crises like the pandemic. Continued research is essential to further understand the dynamics of maternal decision making regarding child health and medication practices, ultimately aiming to promote safer health behaviors among families.

## Figures and Tables

**Table 1 nutrients-16-04182-t001:** General characteristics of participants, Türkiye, 2021.

		Overall *	OTC for Child **	*p*
n		450	204 (45.3)	
Mother’s age, yrs	<30	92 (20.4)	32 (34.8)	0.050
	30–35	204 (45.3)	94 (46.1)	
	>35	154 (34.2)	78 (50.6)	
Father’s age, yrs	<30	39 (8.6)	14 (35.8)	0.410
	30–35	165 (36.6)	74 (44.8)	
	>35	246 (54.6)	116 (47.1)	
Mother’s education, yrs	<12	133 (29.6)	44 (33.1)	0.001
	≥12	317 (70.4)	160 (50.5)	
Father’s education, yrs	<12	165 (36.7)	51 (30.9)	<0.001
	≥12	285 (73.3)	153 (53.7)	
Mother’s occupation	Healthcare worker	71 (15.8)	45 (63.4) ^a^	<0.001
	Teacher	99 (22.0)	46 (46.5) ^b^	
	Other	108 (24.0)	55 (50.9) ^ab^	
	Not working	172 (38.2)	58 (33.7) ^c^	
Father’s occupation	Healthcare worker	30 (6.7)	21 (70.0) ^a^	0.018
	Teacher	52 (11.6)	24 (46.2) ^b^	
	Other	368 (81.7)	157 (43.2) ^b^	
Parent; healthcare worker	At least one	81 (18)	52 (64.2)	<0.001
	None	369 (82)	152 (41.2)	
Family type	Nuclear	372 (82.7)	176 (47.3)	0.182
	Extended	41 (9.1)	15 (36.6)	
	Single mother	37 (8.2)	13 (35.1)	
Family income	Middle–low	219 (48.7)	82 (37.4)	0.001
	High	231 (51.3)	122 (52.8)	
Residence	Semi-urban	215 (47.8)	86 (40.0)	0.030
	City	235 (52.2)	118 (50.2)	
Region	Middle Anatolia	343 (76.2)	154 (44.9)	0.740
	Other	107 (23.8)	50 (46.7)	
**Child characteristics**				
Birth order	1	169 (37.5)	92 (54.4) ^a^	0.003
	≥2	281 (62.4)	112 (39.9) ^b^	
Age, mo	24–47	257 (57.1)	101 (39.3)	0.003
	48–72	193 (42.8)	103 (53.4)	
Sex	Female	205 (45.6)	88 (42.9)	0.348
	Male	245 (54.4)	116 (47.3)	
Gestational age, week	<37	54 (12.0)	26 (48.1)	0.766
	≥37	396 (88.0)	178 (44.9)	
Type of birth	Vaginal birth	153 (34.0)	56 (36.6)	0.008
	Cesarean birth	297 (66.0)	148 (49.8)	
Breastfeeding duration, mo	0–5	59 (13.1)	22 (37.3)	0.308
	6–11	55 (12.2)	31 (56.4)	
	12–17	77 (17.1)	37 (48.1)	
	18–23	158 (35.1)	68 (43.0)	
	≥24	101 (22.4)	46 (45.5)	
Attendance at nursery during pandemic	No	334 (74.2)	130 (38.9)	<0.001
	Yes	116 (25.8)	74 (63.8)	
Presence of any chronic disease	No	415 (92.2)	188 (45.3)	1.000
	Yes	35 (7.8)	16 (45.7)	
Child history of COVID-19 positivity	No	432 (96.0)	191 (44.2)	0.036
	Yes	18 (4.0)	13 (72.2)	
Routine child health supervision during absence of any complaint	No	137 (30.4)	45 (32.8) ^a^	<0.001
	FHU	121 (26.9)	48 (39.7) ^a^	
	Pediatrician	192 (42.6)	111 (57.8) ^b^	
Mother’s perception of child’s weight	Underweight	82 (18.2)	47 (57.3)	0.022
	Normal–overweight	368 (81.8)	157 (42.7)	
Mother’s perception of child’s height	Short	29 (6.4)	18 (62.1)	0.061
	Normal–tall	421 (93.6)	186 (44.2)	
Drug use for child without physician’s recommendation	None	208 (46.2)	76 (36.5) ^a^	<0.001
	Some	194 (43.1)	92 (47.4) ^a^	
	Generally	48 (10.6)	36 (75.0) ^b^	
Maternal self-medication during pandemic	No	251 (55.8)	51 (20.3)	<0.001
	Yes	199 (44.2)	153 (76.9)	
Consideration of OTC medication use after pandemic	No	200 (44.4)	26 (13.0)	<0.001
	Yes	250 (55.6)	178 (71.2)	

* n (%. column percentage); ** n (%. row percentage); ^a^, ^b^, ^c^ Values with different letters were statistically different. *p* < 0.05; OTC: over the counter.

**Table 2 nutrients-16-04182-t002:** Over-the-counter medications for children.

Supplements	n	% *	% **
Overall	204	45.3	100
Vitamin D	102	22.7	50.0
Fish oil	90	20.0	44.1
Propolis	61	13.6	29.9
Multivitamin	60	13.3	29.4
Vitamin C	42	9.3	20.6
Immune booster	33	7.3	16.2
Zinc	28	6.2	13.7
Probiotic	27	6.0	13.2
Herbal tea	25	5.6	12.3
Mouth or nasal spray	18	4.0	8.8
Other	6	1.2	3.0

* In all cases (n = 450); ** in cases that use supplements (n = 204).

**Table 3 nutrients-16-04182-t003:** Reasons for and characteristics of mothers’ use of over-the-counter medications for their children, n = 204.

	n	%
**Purpose for supplementation**		
To prevent disease	165	80.9
To cure the child	69	33.8
To increase intelligence	45	22.1
To increase appetite	41	20.1
To support height growth	14	6.9
**Supplementation period**		
Before illness to protect the child	178	87.3
Post illness to cure the child	36	17.6
Other	11	5.4
**Frequency of supplementation**		
As long as it comes to mind	13	6.4
Once a month	2	1.0
Twice a month	5	2.5
Once or twice a week	41	20.1
Every other day	32	15.7
Every day	109	53.4
**Gave supplements before the pandemic?**		
No	47	23.0
Yes	156	76.5
**Think these supplements are helpful?**		
No	9	4.4
Yes	193	94.6
**The person who provided advice**		
Doctor	167	81.9
Other healthcare worker	73	35.8
Relative/neighbor/teacher	40	19.6
Media	31	15.2
**Fear of side effects**		
None	105	51.5
Some	79	38.7
Mostly or always	20	9.8

**Table 4 nutrients-16-04182-t004:** The relationship between maternal and child characteristics and over-the-counter medication use during the pandemic *.

	AOR	95%Cl	*p*
Mother’s age, yrs	1.1	1.01–1.15	**0.025**
Mother’s education, ≥12 vs. <12 yrs	1.06	0.58–1.95	0.845
At least one parent being healthcare workers, yes vs. no	1.37	0.71–2.63	0.351
Residence, city vs. county	1.11	0.65–1.89	0.693
Family income, high vs. middle low	1.56	0.94–2.59	0.087
Family type,			
Extended vs. nuclear family	1.02	0.40–2.59	0.973
Single mother vs. nuclear family	0.50	0.18–1.38	0.181
Type of birth, Cesarean vs. vaginal	1.71	0.98–2.98	0.059
Birth order, 1 vs. ≥2	2.04	1.12–3.71	**0.019**
Child age, mo, 48–72 vs. 24–47	1.31	0.75–2.28	0.337
Attendance at nursery during pandemic, yes vs. no	2.29	1.22–4.28	**0.010**
Child history of COVID-19 positivity, yes vs. no	3.67	0.87–15.48	0.077
Mother’s perception of child’s nutrition, underweight vs. normal	2.10	1.09–4.05	**0.026**
Drug use for child without physician’s recommendation,			
Some vs. none	1.82	1.05–3.16	**0.032**
Generally vs. none	5.76	2.19–15.17	**<0.001**
Routine child health supervision during absence of any complaint			
FHU vs. none	1.67	0.84–3.34	0.145
Pediatrician vs. none	2.88	1.53–5.41	**0.001**
Maternal self-medication, yes vs. no	12.05	7.13–20.36	**<0.001**
Constant	0.00		0.00

* Multiple logistic regression; AOR: adjusted odds ratio; CI: confidence interval.

## Data Availability

The data presented in this study are available upon request from the corresponding author due to ethical considerations.

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
