# Peer review of "Nutrient Supplements for Young Children and Mothers’ Self Medication with Over-the-Counter Drugs During the COVID-19 Pandemic"

_nutrients, 2024, doi:10.3390/nu16234182_

Round 1
Reviewer 1 Report
Comments and Suggestions for Authors
The article entitled: Nutrient Supplements for Young Children and Mothers’ Self-medication with over-the-counter Drugs during the Covid-19 Pandemic has taken into consideration three medical problems: 1) COVID-19 pandemic, 2) Diet supplementation, 3) OTC drugs. All of the above together describe the dangerous landscape of self-medication. Even though the use of supplements at some point of high processed food consumption can be positive their application without the previous diet and health condition analysis is out of sense and can bring the negative effect. The problem of access to the healthcare system during the COVID-19 pandemic forced people to look for alternative OTC drugs and some kind of physiological "buster". People in critical situations expect magic pills - which resolve their problems. The diet supplements and OTC use in the case of adults are less dangerous than found for children who are in the age of growing period. In this situation higher amount of vitamins, proteins, antioxidants, etc. can influence biochemical processes, leading to defects in basic cellular processes. Moreover, it is well known that OTC in many cases only obscure diseases without a cure effect. Therefore, the unendorsed use of supplements and OTC are potentially dangerous, e.g. high doses of Vit D as a remedy for all health problems.
The article is very well written and readable, the references are correctly cited. The data and statistical analysis are correct. However, the investigated group is too small for the “statistically significant results” and due to that I recommend putting in the title the sentence “preliminary study”.
Secondly, the discussion is too long it looks like a mini-review and due to that should be shortened.
I believe that this article is valuable for people not only scientists from the health care fields.
In conclusion after correction article can be accepted for publication.
Author Response
R1. The article entitled: Nutrient Supplements for Young Children and Mothers’ Self-medication with over-the-counter Drugs during the Covid-19 Pandemic has taken into consideration three medical problems: 1) COVID-19 pandemic, 2) Diet supplementation, 3) OTC drugs. All of the above together describe the dangerous landscape of self-medication. Even though the use of supplements at some point of high processed food consumption can be positive their application without the previous diet and health condition analysis is out of sense and can bring the negative effect. The problem of access to the healthcare system during the COVID-19 pandemic forced people to look for alternative OTC drugs and some kind of physiological "buster". People in critical situations expect magic pills - which resolve their problems. The diet supplements and OTC use in the case of adults are less dangerous than found for children who are in the age of growing period. In this situation higher amount of vitamins, proteins, antioxidants, etc. can influence biochemical processes, leading to defects in basic cellular processes. Moreover, it is well known that OTC in many cases only obscure diseases without a cure effect. Therefore, the unendorsed use of supplements and OTC are potentially dangerous, e.g. high doses of Vit D as a remedy for all health problems. The article is very well written and readable, the references are correctly cited. The data and statistical analysis are correct. However, the investigated group is too small for the “statistically significant results” and due to that I recommend putting in the title the sentence “preliminary study”.
RR1. As seen in method section [Line 113-117 “Using the OpenEpi program[19], it was determined that 384 participants would be needed to calculate a condition with a 50% prevalence at a 5% confidence level, assuming a design effect of 1. Anticipating that 25% of the surveys might be filled out incorrectly, the plan was to collect 480 participants.”], the sample size was calculated using OpenEpi, and since the calculation was based on an assumed 50% prevalence, the maximum possible number of cases was considered. Therefore, we did not regard our study as preliminary and made no changes to the manuscript. However, we plan to conduct further studies with separate calculations for each OTC.
R2. Secondly, the discussion is too long it looks like a mini-review and due to that should be shortened.
RR2. Since the manuscript was prepared as an original article and another reviewer requested additional references for the discussion section, no shortening was made.
R3. I believe that this article is valuable for people not only scientists from the health care fields. In conclusion after correction article can be accepted for publication.
RR3. Thank you for your positive and constructive feedback
Reviewer 2 Report
Comments and Suggestions for Authors
This is an interesting topic of research which is both necessary and important to try and understand more, particularly in light of post-COVID reflections on paediatric healthcare.
I have a few very minor comments. In your excellent and detailed Introduction, it would be useful to mention a little more about the health benefits of breastmilk , particularly in relation to overall paediatric health and immune system development. Duration of breastfeeding is one of the data information topics you focused on, so a stronger thread both in the Introduction and the Discussion / Conclusion, especially as you find that mothers chose supplements to prevent illness, cure the child, increase intelligence, etc - which are all things that breastmilk can achieve. I am interested as to why mothers did not refer back to knowledge about the health benefits of breastmilk, and this may have been due to lack of contact with relevant healthcare professionals during the pandemic who could have re-iterated these strengths.
Can you add how you gained verbal consent from participants in your Method? Was it via phone or a face to face meeting? Can you also confirm clearly what you did with incomplete questionnaires and if you used any data from incomplete questionnaires?
In Table 3 , you mention "lengthening" - some European and American readers won't know what this means , so I suggest using a different term here?
Author Response
R1. This is an interesting topic of research which is both necessary and important to try and understand more, particularly in light of post-COVID reflections on paediatric healthcare. I have a few very minor comments. In your excellent and detailed Introduction, it would be useful to mention a little more about the health benefits of breastmilk , particularly in relation to overall paediatric health and immune system development. Duration of breastfeeding is one of the data information topics you focused on, so a stronger thread both in the Introduction and the Discussion / Conclusion, especially as you find that mothers chose supplements to prevent illness, cure the child, increase intelligence, etc - which are all things that breastmilk can achieve. I am interested as to why mothers did not refer back to knowledge about the health benefits of breastmilk, and this may have been due to lack of contact with relevant healthcare professionals during the pandemic who could have re-iterated these strengths.
RR1. As you mentioned, promoting breastfeeding during pandemics is the most important intervention in supporting children's health. “Breastfeeding is recommended up to two years and beyond, but during pandemic periods and when children are sick, continuing breastfeeding becomes even more cru-cial for supporting child health [6,62,63]. As a limitation, since our study focused on children aged 2-6 years, we did not inquire about their breastfeeding status. In our study, 22.4% of the children were breastfed for 24 months or longer and 36 (28.8%) of the 126 children aged 24-35 months continued breastfeeding after the age of 2. However, the use of OTC medications was 32.0% among breastfed children, compared to 26.7% among those who were not breastfed (p=0.519). This suggests that during the pandemic, the uncertainty surrounding COVID-19 led families to turn to supplementary foods. (line 453-461)” Was added to discussion section.
R2. Can you add how you gained verbal consent from participants in your Method? Was it via phone or a face to face meeting?
RR2. It was corrected as “The survey link was shared via WhatsApp groups where mothers were members and on social media platforms like Instagram using the snowball sampling method. The first page of the Google survey contained an information note and maternal consent. they gave their consent by ticking the option "I agree to participate in the study” Mothers who gave consent proceeded to the second page to answer the questions. (Line 106-110)”
R3. Can you also confirm clearly what you did with incomplete questionnaires and if you used any data from incomplete questionnaires?
RR3. This part was revised for clarity to: "The questionnaires with more than 10% of the questions left partially unanswered (n=6) and those from 36 children who were outside the defined age range were excluded from the study (Lİne 176-179).”
R4. In Table 3 , you mention " lengthening " - some European and American readers won't know what this means , so I suggest using a different term here?
RR4. Instead of 'lengthening,' 'supporting height growth' was used (Table 3).
We thank you for your constructive contributions that have enhanced the quality of our work.